# Environmental parameters of shallow water habitats in the SW Baltic Sea

Markus Franz[1], Christian Lieberum[1], Gesche Bock[1], Rolf Karez[2]

[1] GEOMAR Helmholtz Centre for Ocean Research Kiel, 24105, Germany

[2] State agency for agriculture, environment and rural areas of Schleswig–Holstein, Flintbek, 24220, Germany

*Correspondence to*: Markus Franz (mfranz@geomar.de)

**Abstract.** The coastal waters of the Baltic Sea are subject to high variations in environmental conditions, triggered by natural and anthropogenic causes. Thus, in situ measurements of water parameters can be strategic for our understanding of the dynamics in shallow water habitats. In this study we present the results of a monitoring program at low water depths (1–

10 2.5 m), covering 13 stations along the Baltic coast of Schleswig-Holstein, Germany. The provided dataset consists of biweekly records for dissolved inorganic nutrient concentrations and continuous readings at 10 min interval for temperature, salinity and oxygen content. Data underwent quality control procedures and were flagged respectively. On average, a data availability of >90 % was reached for the monitoring period within 2016–2018. The obtained monitoring data reveal great temporal and spatial variabilities of key environmental factors for shallow water habitats in the southwestern Baltic Sea.

Therefore the presented information could serve as realistic key data for experimental manipulations of environmental parameters as well as for the development of oceanographic, biogeochemical or ecological models. The data associated with this article can be found at https://doi.org/10.1594/PANGAEA.895257 (Franz et al., 2018).

## 1 Introduction

Coastal areas represent highly variable environments. The proximity to land, shallow water depths and the direct influence of

20 river discharges affect oceanography, biology and meteorology of these regions (Sinex, 1994). Drastic changes in water parameters like temperature, oxygen level or pH can occur within hours and minutes, often only detected at local spatial scales (Bates et al., 2018). Marine organisms experiencing these conditions exhibit different strategies to cope with those circumstances. In the context of studies on global change such consideration is crucial, since biological communities are not directly affected by climate itself, but rather by shorter–term variabilities in environmental conditions (i.e. weather)

(Helmuth et al., 2014). However, scientists are still widely applying large–scale averages (temporal and spatial) in experimental approaches, risking misleading interpretations in an either too positive or too negative direction (Bates et al., 2018). This might not always be the result of misconception, but also of limited availability and accessibility of necessary information. Until today, the descriptions of monitoring programs designed for pure data acquisition are commonly published in grey literature and the respective data is not publicly available. The fact that many locations are facing extreme

events that are far beyond average conditions projected for the future (e.g. Mills et al., 2013) underlines the urgent need of fine–scale data. Identifying and describing environmental variabilities (e.g. extreme events) would be a first step towards better predictions of climate change impacts.

The Baltic Sea is a particular example for environmental variability of natural and anthropogenic origin. It exhibits gradients of critical environmental drivers caused by its semi–enclosed characteristics and the large drainage area of surrounding landmasses, mainly reflected in decreasing salinity and temperature towards north (Snoeijs-Leijonmalm et al., 2017). The major contribution to environmental variability caused by human activities in the Baltic Sea is represented by eutrophication. Nutrient concentrations rose until the mid–1980s by riverine inputs of nitrate and phosphate (HELCOM, 2018). The resulting enhanced primary productivity led to an increase of organic matter deposition, which fueled respiration at the seafloor and created hypoxic or even anoxic areas of great spatial extent (HELCOM, 2018). Even though nutrient inputs decreased in the last two decades, so called "dead zones" are persisting, not least because climate warming is boosting deoxygenation (Carstensen et al., 2014). Consequently, shallow areas at the shore can experience episodic hypoxia resulting from oxygen depleted bottom waters and, in addition , by upwelling events that transport water from deeper basins to coastal areas (Conley et al. 2011; Saderne et al., 2013). In the latter case, benthic communities living in these habitats will face not only one, but a set of environmental shifts: besides low oxygen levels, organisms are subjected to increased nutrient concentrations, lower temperatures, higher salinities and elevated $pCO_2$ levels (Lehmann and Myrberg, 2008; Saderne et al., 2013). However, the impact of upwelling events on benthic communities is not straightforward, as the lower temperatures and higher salinities represent short term relaxations from climate driven warming and desalination, being predicted for the Baltic Sea (Jonsson et al., 2018).

The interacting influence of large– and local–scale gradients in water parameters leads to a pronounced variability in environmental conditions of coastal ecosystems in the Baltic Sea. To further develop our understanding of the dynamics communities are experiencing in shallow waters, a monitoring of water parameters along the Baltic coast of Schleswig-Holstein (Germany) was designed. In this contribution we are presenting the obtained data for water temperature, salinity, dissolved oxygen content and nutrient concentrations recorded within the period of 2016–2018.

## 2 Material and Methods

### 2.1 Study area

The monitoring sites are located along the Baltic Sea coast of Schleswig–Holstein, Germany. Thirteen stations were established, with biweekly samplings for dissolved inorganic nutrient concentrations at all stations and continuous recordings of environmental parameters (temperature, salinity and dissolved oxygen) at nine stations, respectively (Fig. 1). The stations are located in boulder field or sandy bottom habitats (Table 1).

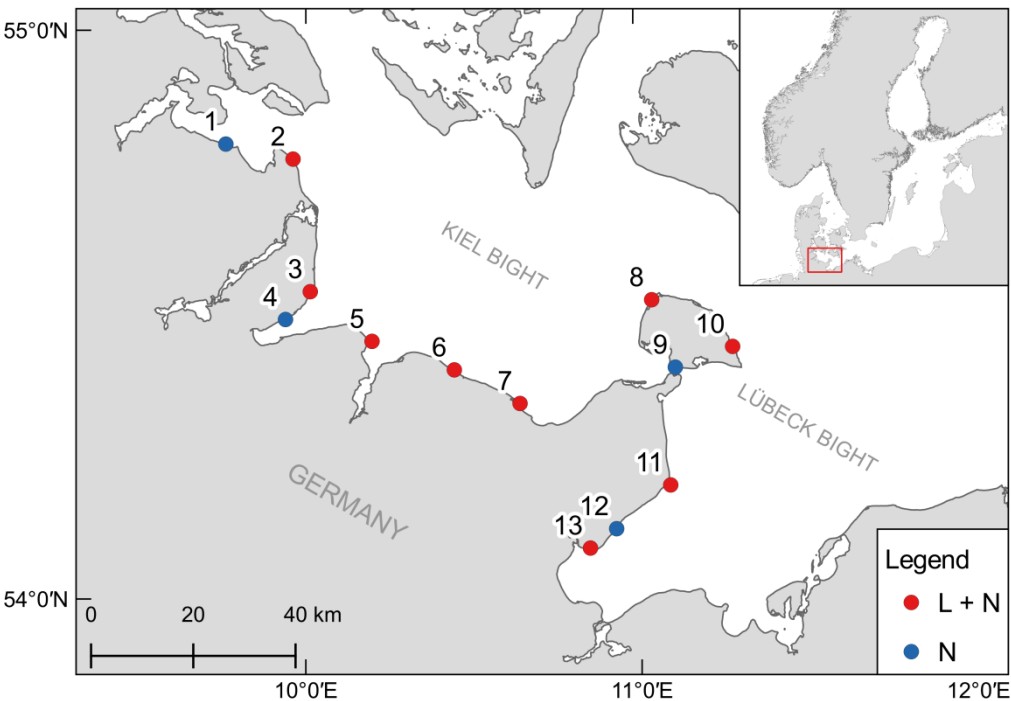

**Figure 1. Geographical position of the 13 monitoring stations. Colors indicate if only samples for dissolved inorganic nutrients were taken (N) or in addition a logger station was deployed (L+N).**

5   **Table 1. Characteristics of the sampling stations in the monitoring program. Station type depicts if only samples for dissolved inorganic nutrients were taken (N) or if in addition continuous recordings by data–loggers (L + N) were performed. Codes for habitat type: BF = Boulder field, SB = Sandy bottom.**

| Station No. | Station type | Latitude | Longitude | Habitat type |
| --- | --- | --- | --- | --- |
| 1 | N | 54.801612 | 9.756799 | SB |
| 2 | L + N | 54.775398 | 9.961210 | BF |
| 3 | L + N | 54.542247 | 10.013464 | BF |
| 4 | N | 54.493296 | 9.938694 | SB |
| 5 | L + N | 54.454751 | 10.199634 | BF |
| 6 | L + N | 54.403929 | 10.447424 | SB |
| 7 | L + N | 54.343969 | 10.644447 | BF |
| 8 | L + N | 54.523752 | 11.044867 | BF |
| 9 | N | 54.404558 | 11.113183 | SB |
| 10 | L + N | 54.439367 | 11.286918 | BF |
| 11 | L + N | 54.197572 | 11.093864 | BF |
| 12 | N | 54.121971 | 10.930081 | SB |
| 13 | L + N | 54.088471 | 10.851146 | BF |

## 2.2 Nutrient sampling and analysis

Twice per month water samples were collected from a water depth of 1 m at all stations. Two water samples were collected at each station per sampling event. In the sampling routine, Stations 1–7 were always sampled in the same week and Stations 8–13 in the following week. Wearing a chest wader, the person collecting the water sample walked until reaching the sampling station at a water depth of approx. 1.2 m. In case of stations equipped with a logger station (see Sect. 2.3), the water sample was collected leaving the shore line in a roughly perpendicular direction heading towards the logger station until reaching the desired water depth of approx. 1.2 m. Water samples were collected using a 50 mL syringe connected to a plastic tube of 1 m length. For sampling, the syringe was held at the water surface and the tube was lowered to 1 m depth. Care was taken not to suck sediment from the sea floor. All sampling equipment (syringe, tube, scintillation vials) was rinsed with water from the collection site before the actual sample was taken. Immediately after the water was collected with the syringe, it was filtered using a cellulose acetate syringe filter (0.45 µm) while filling the sample into a scintillation vial. Back in the laboratory, the samples were directly put into a freezer and kept at -20 °C until further sample processing. Subsequently, the samples were analyzed for the concentration of dissolved inorganic nutrients (total oxidized nitrogen, nitrite, ammonia, phosphate and silicate) by UV/VIS spectroscopy using a Continuous Flow Analyzer (San++ Automated Wet Chemistry Analyzer, Skalar Analytical B.V.). For the analyses, respective chemical methods provided by Skalar were applied (Table 2). Total oxidized nitrogen concentrations ($NO_x$) were determined by the cadmium reduction method, followed by measurement of nitrite (originally present plus reduced nitrite).

**Table 2. Chemical methods applied to measure dissolved inorganic nutrient concentrations in the water samples. NOx: Total oxidized nitrogen.**

| Dissolved inorganic nutrient | Skalar chemical method no. | References |
|---|---|---|
| $NO_x$ | 461 | Greenberg et al., 1980; Walinga et al., 1989; Navone, 1964; ISO 13395, 1996 |
| Nitrite | 467 | EPA, 1974; Greenberg et al., 1980; ISO 3696, 1987; ISO 13395, 1996 |
| Ammonia | 156 | Krom, 1980; Searle, 1984; ISO 3696, 1987 |
| Phosphate | 503 | Boltz and Mellon, 1948; Greenberg et al., 1980; ISO 3696, 1987; Walinga et al., 1989; ISO 15681-2, 2003 |
| Silicate | 563 | Babulak and Gildenberg, 1973; Smith and Milne, 1981; ISO 3696, 1987; ISO 16264, 2002 |

## 2.3 Data logger setup

Nine logger stations were deployed in the field at a depth of 2.5 m to continuously record data for temperature, salinity and dissolved oxygen. Each logger station consisted of a concrete slab (50 x 50 cm) equipped with a vertical threaded stainless steel bar, which was used as a mounting structure for data loggers (Fig. 2). The data loggers were fixed at the threaded

stainless steel bar 40 cm above the seafloor. To protect the sensors from fishing gear and drifting material, a frame was constructed around the setup by connecting plastic tubes to the end of the metal rod and to each side of the concrete slab.

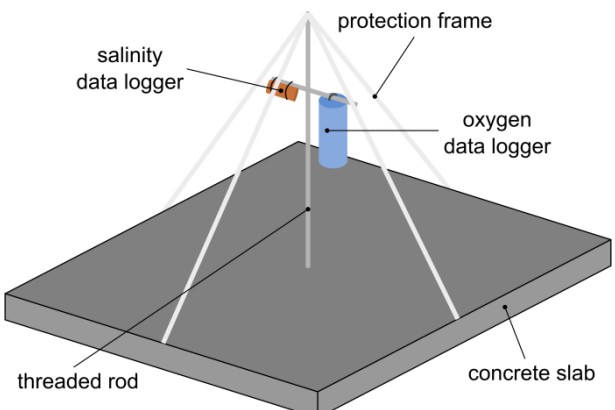

**Figure 2. Scheme of deployed logger station**

Two types of self–contained data loggers were used for the logger stations: (I) MiniDOT loggers (Precision Measurement Engineering; http://pme.com; $\pm10$ µmol L$^{-1}$ or $\pm5$ % saturation) including antifouling copper option (copper plate and mesh) to measure dissolved oxygen concentration and (II) DST CT salinity & temperature loggers (Star–Oddi; http://star-oddi.com; $\pm1.5$ mS cm$^{-1}$) recorded the conductivity. Both sensors additionally recorded water temperature with an accuracy of $\pm0.1$ °C.

10    The sampling interval was set to 10 minutes for all parameters. Date and time were saved in UTC. All loggers were pre–calibrated by the manufacturer. The fully prepared logger stations were deployed at the sites by SCUBA diving. This resulted in differing starting dates, as the deployment was depending on weather conditions and staff availability. For read–out the loggers were detached from the logger station by SCUBA divers and connected to a laptop on land (Table 3). The loggers were cleaned from fouling organisms during read–outs and on an irregular basis during summer and autumn of each

15    year, because fouling is expected to be highest in these seasons. However, as fouling could not be avoided at all times, quality control procedures were applied to the recorded data in order to identify sensor drifts (see Sect. 2.4.2).

**Table 3. Read–out dates of self–contained data loggers at the respective stations. Asterisk indicates when only loggers for dissolved oxygen concentration were read out.**

| Station No. | 1st read–out | 2nd read–out | 3rd read–out | 4th read–out | 5th read–out |
|---|---|---|---|---|---|
| 2 | 25 Apr 2016 | 11 Nov 2016 | 03 Mar 2017 | 15 Nov 2017 | 23 May 2018 |
| 3 | 25 Apr 2016* | 14 Dec 2016 | 02 Feb 2017 | 19 Oct 2017 | 03 Apr 2018 |
| 5 | 26 Apr 2016* | 12 Aug 2016 | 03 Mar 2017 | 19 Oct 2017 | 04 Apr 2018 |
| 6 | 30 Mar 2016 | 26 Apr 2016 | 12 Aug 2016 | 15 Aug 2017 | – |
| 7 | 26 Apr 2016 | 19 Aug 2016 | 30 Mar 2017 | 15 Aug 2017 | 27 Mar 2017 |
| 8 | 27 Apr 2016 | 29 Sep 2016 | 17 May 2017 | 23 Aug 2017 | – |
| 10 | 27 Apr 2016 | 02 Nov 2016 | 17 May 2017 | 17 Aug 2017 | 17 May 2018 |
| 11 | 27 Apr 2016 | – | 16 Feb 2017 | 26 Oct 2017 | 16 May 2018 |
| 13 | 27 Apr 2016 | 16 Nov 2016 | 16 Feb 2017 | 26 Oct 2017 | 16 May 2018 |

## 2.4 Data processing

### 2.4.1 Dissolved inorganic nutrients

5    To calculate the concentration of nitrate in the sample, the nitrite concentration was subtracted from the concentration of $NO_x$.

### 2.4.2 Temperature, salinity and dissolved oxygen

The data for dissolved oxygen (DO) concentration were corrected for a depth of 2.5 m using the software provided by the manufacturer. Additionally, a manual compensation for salinity was calculated according to Eq. (1):

$$C_{comp} = C_m * \frac{e^{C_{eS}}}{e^{C_{eF}}} \tag{1}$$

where $C_m$ is the measured DO content. $C_{eF}$ and $C_{eS}$ are the calculated saturation equilibrium concentration in freshwater and saltwater, respectively. The calculation was done according to Garcia and Gordon (1992) (constants used from Benson and Krause Jr (1984)) using temperature and salinity measured by the data loggers at the same time point.

15    Negative readings in the databases of temperature, salinity and DO concentration were removed, as well as values recorded at the day of read–out. Quality control was carried out by implementing spike and gradient tests according to the SeaDataNet quality control manual (https://seadatanet.org). The spike test identifies differences in sequential measurements and was calculated according to Eq. (2):

$$test\ value = \left| x_i - \left( \frac{x_{i+1} + x_{i-1}}{2} \right) \right| - \left| \left( \frac{x_{i+1} + x_{i-1}}{2} \right) \right| \tag{2}$$

where $x_i$ is the actual measurement, $x_{i-1}$ and $x_{i+1}$ are the previous and next values in the record sequence, respectively. The gradient test identifies transitions of adjacent values that are too steep. The calculation was carried out following Eq. (3):

$$test\ value = \left|\left(\frac{x_{i+1} + x_{i-1}}{2}\right)\right| \tag{3}$$

Threshold values for both tests were $\geq 0.5$ °C for temperature data, $\geq 1$ for salinity measurements and $\geq 0.5$ mg L$^{-1}$ for DO content records. Data that passed both tests were, in addition, visually inspected. Here, emphasis was put on erroneous readings resulting from biofouling of the sensors. Optical sensors typically show drift towards saturation. Conductivity sensors tend to steadily decrease, if the sensor is being fouled (Garel and Ferreira, 2015). Therefore, the complete dataset for both sensor types were plotted and inspected. The measurements of DO concentration were checked for extended periods of

full saturation, often visible as plateaus in the plots. For the salinity data, the visual inspection focused on continuous declines followed by immediate returns to measured values before the decrease started. Dates of read–outs and sensor cleaning were additionally used to support the identification of sensor drift. All data values were flagged according to applied quality checks (Table 4).

In order to exemplify spatial and temporal differences in the overall variability of temperature, salinity and dissolved oxygen

concentration, the complete datasets obtained for Stations 2 and 13 were plotted.

**Table 4. Data quality flags assigned to records of temperature, salinity and dissolved oxygen concentration. Flags are based on quality checks by spike test, gradient test and visual inspection.**

| Flag | Name | Description |
|------|------|-------------|
| 1 | Pass | Data value that passed all applied quality checks |
| 2 | Suspect | Data value that failed either in spike test or in gradient test |
| 3 | Fail | Data value that failed both in spike test and gradient test |
| 4 | Visually suspect | Data value identified as erroneous reading by visual inspection |
| 5 | Salinity compensation fail | Data value for dissolved oxygen concentration that was not compensated for salinity |

**3. Data availability**

All datasets are deposited as a collection at PANGAEA and can be accessed via the following DOI: https://doi.org/10.1594/PANGAEA.895257 (Franz et al., 2018).

## 3.1 Dissolved inorganic nutrients

Data for dissolved inorganic nutrient concentrations are available from 01 February 2016 to 26 March 2018 (Fig. 3). Gaps in the data for nutrients result from missed sampling events due to staff unavailability. The quantity of available data points in the monitoring period ranges from 51 to 55 out of 56 samples that could have been taken per station. Thus, on average 93 % of the potential full data coverage was reached. The information on dissolved inorganic nutrient concentrations is organized in one data file summarizing the results of all monitoring stations.

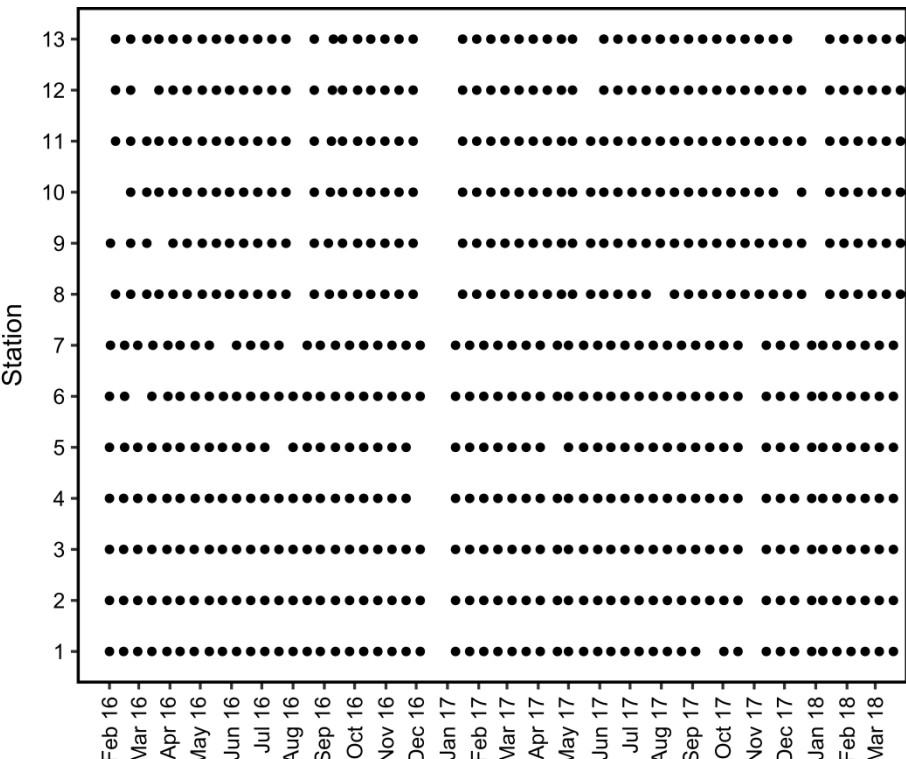

**Figure 3. Data availability of dissolved inorganic nutrient concentrations. Each dot represents a measurement for total oxidized nitrogen, nitrite, ammonia, phosphate and silicate.**

## 3.2 Temperature, salinity and dissolved oxygen

Records of temperature, salinity and DO concentration are available from 16 February 2016 to 23 May 2018 (Fig. 4). Missing data resulted from read–outs and sensor malfunctions (Table 3). For Station 6 no data from the logger station is available due to complete failure of the deployed data loggers. Datasets of Stations 3, 5, 8 and 11 end in 2017, as the logger stations were not found back at the last read–out date (Table 3), probably due to sedimentation processes. On average, 96 % of records for temperature, salinity and DO concentration were retained from raw dataset after data processing (see Sect.

2.4.2) and a mean data availability of 671 days (1.8 years) was reached. On average, more data is available for salinity and temperature (690 days) than for DO content (653 days). For each station and logger type a single data file is given at PANGAEA.

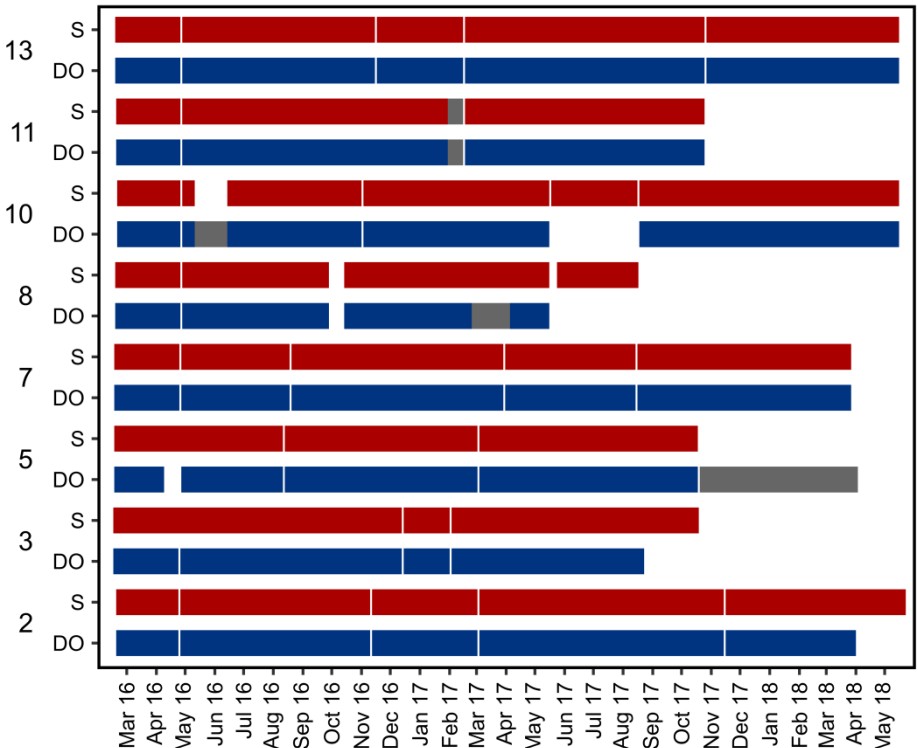

Figure 4. Data availability of dissolved oxygen (DO) concentration and salinity measured by self–contained data loggers. Both logger types (DO and salinity) additionally measured temperature. Temperature data is always available for the same periods as DO and salinity, respectively. Grey bars indicate cases where only temperature data is available. Y–axis shows station numbers and respective logger type. Note that only data with quality flag = 1 are considered.

## 4. Data overview

### 4.1 Dissolved inorganic nutrients

The measurements of nitrate concentrations exhibited differences between monitoring stations. Interquartile ranges (IQR) were higher for Stations 11–13 than for the remaining ones (Fig. 5a). All stations showed numerous outliers (values > 1.5 times IQR), with highest values measured for Station 13. Overall nitrate concentrations range between 0–121.41 µmol L$^{-1}$ (Table 5).

The records for nitrite concentrations show larger IQR in general. Here, Stations 1 and 4 as well as 11–13 are found to be more variable than the other stations (Fig. 5b). The outliers are evenly distributed over all stations. Nitrite concentrations range between 0–1.04 µmol L$^{-1}$, with highest measurements recorded for Station 13 (Table 5).

The measured concentrations for ammonia show a similar variability as the records for nitrate concentrations. Highest IQR are again found for Station 11–13. Moreover, Station 13 exhibited the highest ammonia concentrations measured (Fig. 5c). Concentrations of ammonia range between 0.43–17.46 µmol $L^{-1}$ (Table 5).

The phosphate concentrations show low variability over all stations (Fig. 5d). Among all dissolved inorganic nutrients measured, phosphate reveals the lowest number of outliers, with Stations 4, 7, 8, 10 and 11 being without outliers at all. The highest concentration of phosphate was recorded for Station 3. In total, concentrations range between 0.04–3.39 µmol $L^{-1}$ (Table 5).

The measured silicate concentrations display comparable variabilities like the measurements for phosphate concentrations. Stations 1 and 4 as well as 11–13 feature slightly larger IQR than the other stations (Fig. 5e). Noticeably, Stations 7–9 exhibit no outliers. At Station 13, the highest concentration of silicate was registered. The data for silicate concentrations ranges between 0–67.66 µmol $L^{-1}$ (Table 5).

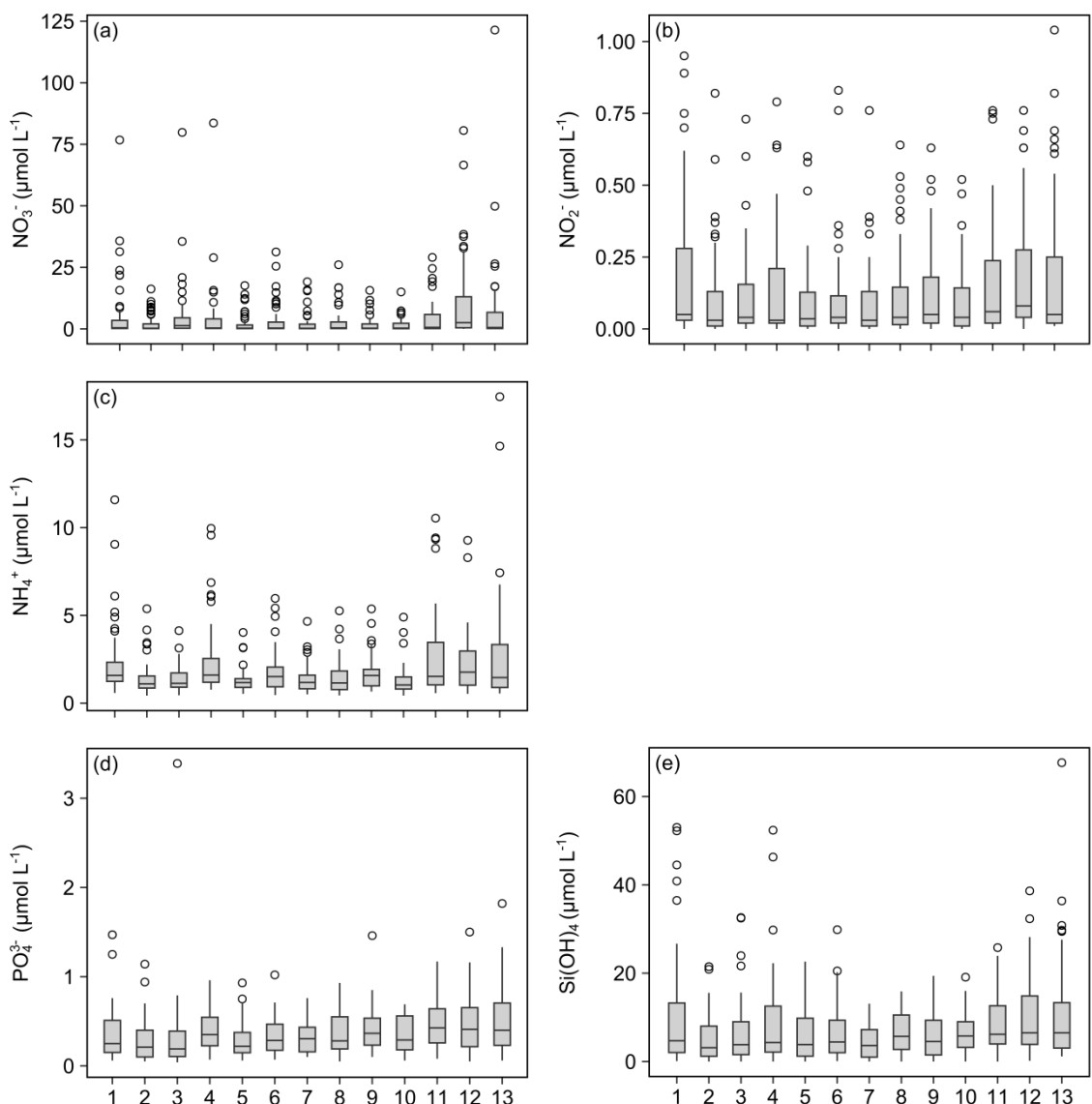

**Figure 5. Boxplots of measured dissolved inorganic nutrient concentrations. Data for nitrate (a), nitrite (b), ammonia (c), phosphate (d) and silicate (e) concentrations are presented. X–axis indicates station numbers. Boxes indicate median, first and third quartile. Whiskers show values within and dots values outside 1.5–times inter–quartile range.**

**Table 5. Summary of measured dissolved inorganic nutrient concentrations in water samples from the monitoring stations. For each station and nutrient, median, minimum and maximum concentrations are presented.**

| Station | $NO_3^-$ ($\mu$mol L$^{-1}$) | $NO_2^-$ ($\mu$mol L$^{-1}$) | $NH_4^+$ ($\mu$mol L$^{-1}$) | $PO_4^{3-}$ ($\mu$mol L$^{-1}$) | $Si(OH)_4$ ($\mu$mol L$^{-1}$) |
|---|---|---|---|---|---|
| 1 | 0.45; 0.00; 76.75 | 0.05; 0.00; 0.95 | 1.58; 0.58; 11.59 | 0.25; 0.06; 1.47 | 4.70; 0.09; 53.02 |
| 2 | 0.25; 0.00; 16.20 | 0.03; 0.00; 0.82 | 1.10; 0.43; 5.38 | 0.21; 0.05; 1.14 | 3.13; 0.00; 21.49 |
| 3 | 1.30; 0.01; 79.84 | 0.04; 0.00; 0.73 | 1.13; 0.45; 4.13 | 0.19; 0.04; 3.39 | 3.18; 0.00; 32.59 |
| 4 | 0.35; 0.02; 83.63 | 0.03; 0.00; 0.79 | 1.60; 0.77; 9.96 | 0.35; 0.07; 0.96 | 4.30; 0.00; 52.37 |
| 5 | 0.29; 0.00; 17.61 | 0.04; 0.00; 0.60 | 1.18; 0.53; 4.03 | 0.22; 0.06; 0.93 | 3.84; 0.00; 22.62 |
| 6 | 0.33; 0.01; 31.26 | 0.04; 0.00; 0.83 | 1.52; 0.46; 5.97 | 0.29; 0.07; 1.02 | 4.43; 0.09; 29.84 |
| 7 | 0.24; 0.00; 19.14 | 0.03; 0.00; 0.76 | 1.18; 0.50; 4.66 | 0.31; 0.10; 0.76 | 3.62; 0.00; 13.10 |
| 8 | 0.40; 0.00; 26.06 | 0.04; 0.00; 0.64 | 1.15; 0.44; 5.26 | 0.28; 0.05; 0.93 | 5.70; 0.00; 15.88 |
| 9 | 0.27; 0.00; 15.64 | 0.05; 0.00; 0.63 | 1.58; 0.66; 5.37 | 0.37; 0.10; 1.46 | 4.53; 0.00; 19.40 |
| 10 | 0.31; 0.02; 15.02 | 0.04; 0.00; 0.52 | 1.04; 0.43; 4.90 | 0.29; 0.06; 0.69 | 5.79; 0.00; 19.10 |
| 11 | 0.61; 0.01; 29.04 | 0.06; 0.00; 0.76 | 1.53; 0.57; 10.54 | 0.43; 0.08; 1.17 | 6.17; 0.00; 25.80 |
| 12 | 2.55; 0.02; 80.58 | 0.08; 0.00; 0.76 | 1.77; 0.53; 9.28 | 0.41; 0.05; 1.50 | 6.50; 0.17; 38.63 |
| 13 | 0.55; 0.03; 121.41 | 0.05; 0.01; 1.04 | 1.46; 0.55; 17.46 | 0.40; 0.06; 1.82 | 6.50; 1.16; 67.66 |

## 4.2 Temperature, salinity and dissolved oxygen

Temperature data show little variation between the different monitoring stations, indicated by largely overlapping boxplots (Fig. 6a, c). The records range between 0–22 °C (Table 6), with a maximum daily change in temperature of 8 °C in May 2017 (Station 13). As a result of a shorter monitoring period (Fig. 4), medians of Stations 3, 5 and 11 tend to be higher than the remaining ones. The records of temperature at these stations end in October 2017 and therefore the medians were not influenced by low temperatures in winter 2017 and spring 2018.

Data indicate a decrease in salinity of 0.02 per km (regression: $y = 14 - 0.22x$, p = 0.013) from Station 2 to Station 13 (Fig. 6b). This represents an overall decrease in salinity of 3.8 over a straight–line distance of 174 km. Variability among the stations is very similar, overall values range between salinities of 3–22 (Table 6). The largest fluctuation in salinity within a single day was recorded in March 2017 at Station 13 with a change of 9. Highest measurements were recorded for Stations 3 and 7, lowest for Station 8.

Measurements of DO concentration showed consistently similar medians for all stations (Fig. 6d). Noticeably, the variability among the stations differed. Data of Station 8 ranges between 6–15 mg L$^{-1}$, while data of Station 13 varies between 0–21 mg L$^{-1}$ (Table 6). For the latter, the largest daily range of 15 mg L$^{-1}$ was recorded in August 2016.

The exemplary detailed comparison of Stations 2 and 13 reveals substantial differences in variability and seasonal dynamics of the measured variables (Fig. 7). Temperature, being the least fluctuating measurement, shows no apparent differences in its overall trends (Fig. 7a). However, the variability in summer months tends to be more pronounced at Station 13, especially visible in summer 2017. The salinity records for both stations do not follow a seasonal pattern, e.g. showing steep increases in summer 2016 as well as winter 2017 (Station 2, Fig. 7b). Noticeably, shorter term fluctuations (within a month) appear to be more common at Station 13, being especially strong in spring to summer 2017. The data for DO concentration show great

differences in the degree of short term variability among the two compared stations (Fig. 7c). Here, Station 2 is again less variable, displaying a more compressed trend. In contrast, DO concentrations at Station 13 vary strongly within short time frames (few days) with differences of up to ~15 mg L$^{-1}$ , e.g. in summer 2017 (Fig. 7c). Besides spatial differences in variability, fluctuations of DO concentrations are generally amplified in spring to autumn at both stations.

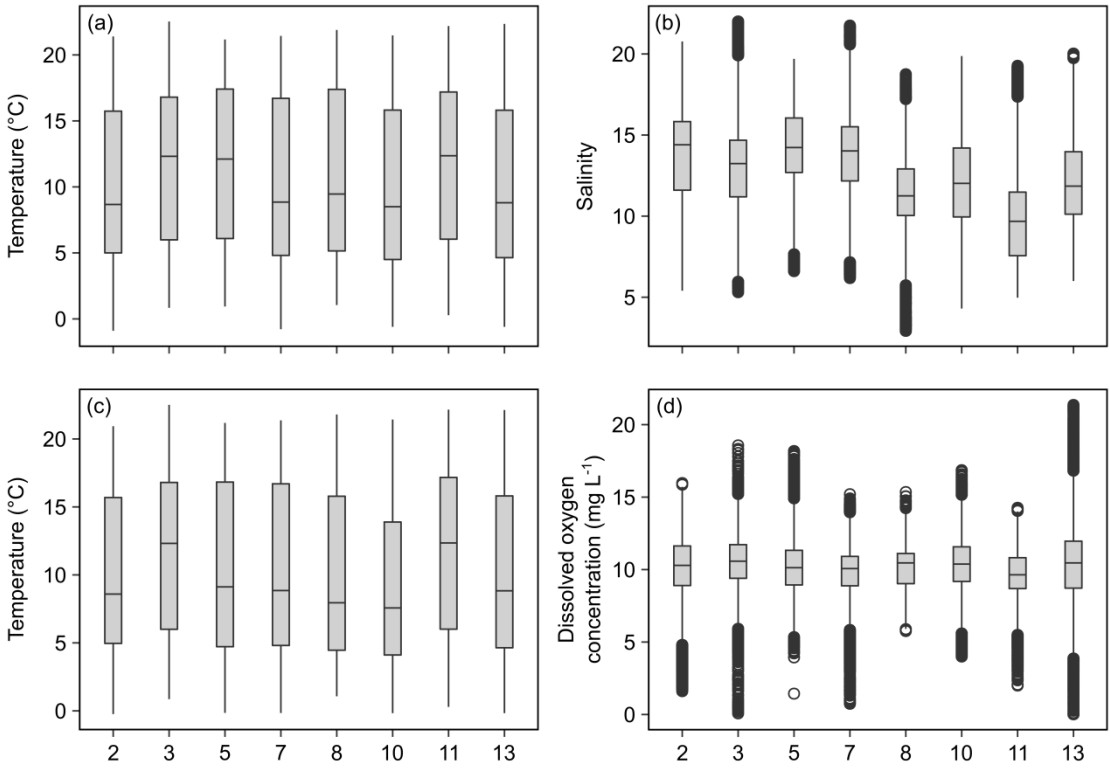

**Figure 6. Boxplots of temperature, salinity and dissolved oxygen concentration recorded over the monitoring period by DST CT (a, b) and MiniDOT (c, d) data loggers, respectively. X–axis indicates station numbers. Boxes indicate median, first and third quartile. Whiskers show values within and dots values outside 1.5–times inter–quartile range. Note that only data with quality flag = 1 were plotted.**

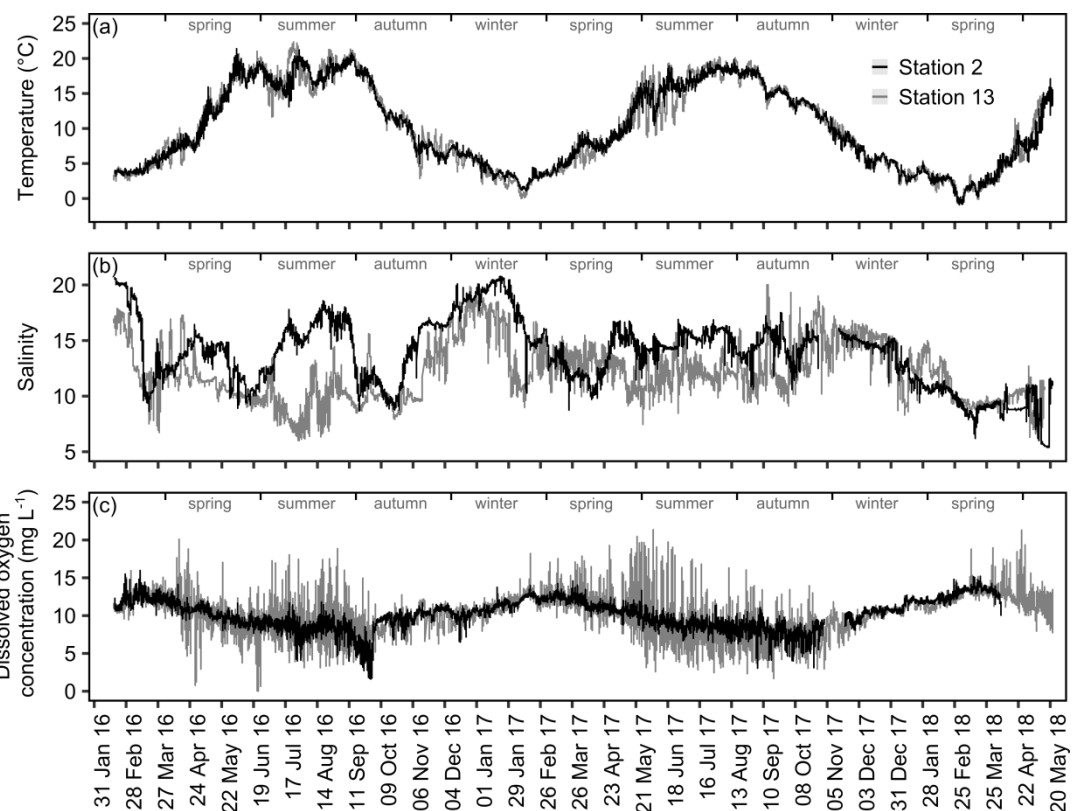

**Figure 7. Exemplary overview of temperature (a), salinity (b) and dissolved oxygen concentration (c) datasets for Stations 2 and 13. The presented data were recorded by DST CT (a, b) and MiniDOT (c) data loggers, respectively. Note that only data with quality flag = 1 were plotted.**

**Table 6. Median and range of recorded measurements obtained by self–contained data loggers over the respective monitoring period. Temperature (T), dissolved oxygen concentration (DO) and salinity (S) were measured by two types of data loggers. Note that data can cover different time periods (see Sect. 3.2) and that only data with quality flag = 1 were considered.**

| Station | Logger type | Parameter | Median | Minimum | Maximum |
|---|---|---|---|---|---|
| 2 | MiniDOT | T (°C) | 8.59 | -0.24 | 20.94 |
| | | DO (mg L$^{-1}$) | 10.29 | 1.61 | 15.97 |
| | DST CT | T (°C) | 8.67 | -0.90 | 21.40 |
| | | S | 14.40 | 5.40 | 20.77 |
| 3 | MiniDOT | T (°C) | 12.33 | 0.86 | 22.51 |
| | | DO (mg L$^{-1}$) | 10.58 | 0.08 | 18.57 |
| | DST CT | T (°C) | 12.32 | 0.84 | 22.53 |
| | | S | 13.24 | 5.31 | 22.00 |
| 5 | MiniDOT | T (°C) | 9.12 | -0.15 | 21.18 |
| | | DO (mg L$^{-1}$) | 10.13 | 1.44 | 18.18 |
| | DST CT | T (°C) | 12.11 | 0.94 | 21.17 |
| | | S | 14.23 | 6.61 | 19.70 |
| 7 | MiniDOT | T (°C) | 8.85 | -0.16 | 21.37 |
| | | DO (mg L$^{-1}$) | 10.07 | 0.74 | 15.20 |
| | DST CT | T (°C) | 8.85 | -0.78 | 21.44 |
| | | S | 14.02 | 6.18 | 21.75 |
| 8 | MiniDOT | T (°C) | 7.95 | 1.07 | 21.80 |
| | | DO (mg L$^{-1}$) | 10.46 | 5.76 | 15.35 |
| | DST CT | T (°C) | 9.46 | 1.05 | 21.89 |
| | | S | 11.25 | 2.93 | 18.75 |
| 10 | MiniDOT | T (°C) | 7.57 | -0.17 | 21.43 |
| | | DO (mg L$^{-1}$) | 10.38 | 4.00 | 16.86 |
| | DST CT | T (°C) | 8.50 | -0.6 | 21.48 |
| | | S | 12.02 | 4.30 | 19.87 |
| 11 | MiniDOT | T (°C) | 12.35 | 0.29 | 22.16 |
| | | DO (mg L$^{-1}$) | 9.64 | 2.00 | 14.26 |
| | DST CT | T (°C) | 12.37 | 0.28 | 22.19 |
| | | S | 9.68 | 4.97 | 19.27 |
| 13 | MiniDOT | T (°C) | 8.83 | -0.17 | 22.13 |
| | | DO (mg L$^{-1}$) | 10.46 | 0.02 | 21.36 |
| | DST CT | T (°C) | 8.80 | -0.60 | 22.35 |
| | | S | 11.85 | 6.00 | 20.02 |

# 5. Conclusions

The present environmental monitoring provides bimonthly data for dissolved inorganic nutrient contents and continuous records of temperature, salinity and DO concentration. The obtained data provide a good picture of the highly dynamic shallow waters at the Baltic coast of Schleswig-Holstein, Germany. The total range (Fig. 6) as well as the recorded maximum diurnal changes (see Sect. 4.2) of continuously measured parameters highlight the enormous environmental fluctuations biota are facing in these habitats. However, the closer examination of data obtained for Stations 2 and 13, which were exemplarily depicted, shows that the extent of environmental variability locally depends on spatial and temporal factors (Fig. 7). This is especially evident for measurements of salinity and DO concentration. At both stations, the trends in salinity neither followed a recurring seasonal pattern nor was there a systematic difference between the stations, which could be expected according to the results of the applied regression using median salinities of the overall monitoring period (see Sect. 4.2). In contrast, the observed fluctuations have different timings at the stations, indicating that local influences like rain water discharges could co-occur with large scale processes, e.g. inflow events of North Atlantic waters (Snoeijs-Leijonmalm et al., 2017, Reusch et al., 2018). Local factors might also explain the rapid increase in salinity within a single day at Station 13 that was paralleled by DO concentrations dropping to 1.7 mg L$^{-1}$, an indication of oxygen depleted water being upwelled at the coast. The variability of measured DO concentrations also exhibits local differences and, in addition, seasonal patterns. The overall stronger variability of DO data at Station 13 could be related to higher nutrient loads compared to Station 2. The median nitrate concentrations at this station were 2–times higher, the recorded maximum exceeded even 7.5–times the value of Station 2 (Table 5). In consequence, biological productivity could be enhanced at this station, leading to larger diurnal fluctuations (Gubelit and Berezina, 2010), e.g. recorded as a drop by 50% in DO concentration in August 2016. The generally lower variability in winter at both stations could be mediated by lower temperatures, since biological processes are slowed down in this period, keeping oxygen concentrations more stable. The likelihood of detecting such events and processes by repeated single measurements is very low, thus continuous records provide the chance to unravel short term dynamics that might have not been described by common monitoring efforts.

The bimonthly measured nutrient loads exhibited seasonal fluctuations with highest concentrations in autumn and winter (Fig. 5). Noticeably, the present extremes in nitrate concentrations exceed 6.6–times the highest recorded concentration measured at a close–by time series station between 1957–2014 at a depth of 1 m (http://bokniseck.de/; Lennartz et al., 2014; Bange and Malien, 2015). The Boknis Eck time series station is situated 2.2 km away from Station 3, but located further offshore. Therefore, the time series station could be less affected by local river runoff and groundwater seepage that might explain the increased nitrate loads in water samples of this study (Szymczycha and Pempkowiak, 2016). Indeed, groundwater seepage has been described as a significant pathway in the hydrology of Eckernförde Bay (Schlüter et al., 2004), a location that is covered in this monitoring and where measured nitrate concentrations were among the highest (Stations 3 and 4, see Table 5).

The employment of self–contained monitoring systems, as presented here, always poses the risk of data losses due to failure of data loggers. Since the used data loggers have to be read out manually, a malfunction could be detected with temporal delay, leading to substantial gaps in the records. Real time data systems like measurement buoys provide the advantage of wireless data transmission, allowing a continuous control of functionality. Furthermore, sensors attached to buoys are less

susceptible to sediment dynamics. The coverage of the logger station by sand potentially will lead to extended gaps in coming datasets, since four of the setups were not found back during the last read–out. Nevertheless, the advantages of real time data systems go along with an increase in expenses for single measurement systems by up to an order of magnitude. Therefore, in this study, independent data loggers have been applied, keeping costs lower and favoring a higher replication of the measurement stations. The average retained amount of data (96 %) from the raw dataset of temperature, salinity and

DO measurements shows that this strategy can be a worthwhile alternative to more expensive monitoring systems. For future deployments the geological characteristics of the sampling sites should be examined more in detail, e.g. to avoid areas of pronounced sediment transport.

The obtained, temporally fine scaled data of this study could be utilized for diverse purposes. The records could support the development and skill assessment of models (oceanographic, biogeochemical and ecological) or provide background

information for the definition and probability of extreme events like heatwaves or coastal upwelling (Bennett et al., 2013; Hobday et al., 2016; Bates et al., 2018). It may serve as a base for more realistic experimental approaches that apply not only mean treatment levels but also consider the range and magnitude of environmental variation (Wernberg et al., 2012). Moreover, the data could help to identify mechanisms behind observed changes in biological monitoring programs performed in the same area. Since the assessment will be continued for the coming years, the temporal coverage of the

dataset is going to broaden in the future and may even enhance its applicability.

Versatile possible uses of the presented data underline the great value of observational programs. As the human population is concentrated along the coastlines, signals of anthropogenic changes are expected to be particularly strong. However, the detection of change in coastal areas is a challenging task, since human pressures and natural variability caused by the sea–land interface vastly overlap (Cloern et al., 2016). At this point, environmental data of high resolution can be a useful tool, as

they allow a better differentiation of natural variability from sings of human impact. Traditional monitoring strategies (e.g. ship–based surveys) can usually not provide the necessary information, since temporal resolution is low and access to shallow waters limited. Self–contained monitoring systems therefore appear as a suitable alternative to bridge this knowledge gap. The presented approach displays not only an opportunity to researchers in better understanding the system dynamics, but also for policymakers in choosing appropriate measures in future environmental protection (Helmuth et al.,

2014).

**Author contributions**

MF wrote the manuscript and processed the raw data. CL and GB performed maintenance of the logger stations as well as samplings for nutrient analyses. RK initiated and designed the monitoring program. All authors participated in data logger read–outs.

**Competing interests**

The authors declare that they have no conflict of interest.

**Acknowledgements**

We would like to thank Vanessa Herhoffer for her conscientious work collecting water samples, Bente Gardeler for nutrient analyses and various divers for help in finding and maintaining logger stations. MF acknowledges the financial support by the state agency for agriculture, environment and rural areas of Schleswig–Holstein (LLUR; reference number: 0608.451426).

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
