# Peer review of "Environmental parameters of shallow water habitats in the SW Baltic Sea"

_Earth System Science Data, 2018_

## Referee Comment (RC1) · Anonymous Referee #1 · 9 Feb 2019

General comments The manuscript is well structured and written. The collection and preparation of data has also been done in a mostly satisfying way. However, the study suffers from a couple of major issues: A) In my opinion, the novelty of the study applies, at the most, to the southern Baltic. At least regarding the northern parts of the Baltic (Sweden, Finland, Estonia), corresponding data from shallow, coastal environments can be found in a large number of reports and publications. Albeit those data are often considered trivial and have thus not been highlighted and summarized as systematically as in the manuscript. In any case, the authors should know that these kind of data do exist, use them for comparison and refer to the sources accordingly. Instead of a case study, there would rather be need for a systematic review of data from shallow, coastal areas of the Baltic. B) Although the manuscript is supposed to be of descriptive

nature, a proper synthesis of the findings is lacking. This would be especially important in a study without a specific theoretical framework or hypothesis. C) One of the main points of the study is to highlight the fact that shallow, coastal environments are highly dynamic. However, this is hardly surprising. At least I would not expect anything else than high diurnal and seasonal variation in most environmental variables from shallow, brackish waters located between the 54th and 55th latitudes. While the authors highlight the dynamism of the surveyed environmental type, they don't summarize its (diurnal and seasonal) dynamics in any kind of way. A spatial comparison of the dynamics and potential phenological differences among the stations would be a major improvement in the manuscript. This would be possible with the existing data. D) In order to highlight the dynamic nature of the studied environments, they should be compared with something. First, the authors should refer to corresponding data from other shallow environments (see point A). More importantly, the set-up of the study should have included deeper waters further off the shore. Preferably, the study set-up should have incorporated coastal gradients (inshore-offshore) by each station. Although it is understandable that this kind of an approach would have required completely different resources, highlighting the dynamism of the studied environment without actually comparing it with anything is, in my opinion, a major issue.

Specific comments Page 2, rows 12-15: Benthic communities will face not only low or high extremes but high variation in environmental conditions. In shallow areas, both sorts of extremes are very common.

Page 2, rows 16-18: At least for the northern Baltic, there is already plenty of reports and scientific publications where this type of data can be found. However, those data may be considered too trivial to be summarized in their own right. In other words, the systematic summary presented in this manuscript can be considered useful (but not without references to and comparisons with data from other shallow, coastal areas of the Baltic).

Page 2, rows 23-25: As the purpose of the study is to show that shallow, nearshore

habitats are particularly dynamic compared with deeper, off-shore habitats, depth gradients should absolutely be included in the sampling set-up (see also general comments).

Page 4, rows 4-9: Wading to the sampling stations can have had large effects on many of the measured variables because of sediment resuspension. The authors should be able to show that such effects did not take place. In my opinion, wading should absolutely be avoided, preferring other ways of approaching the stations.

Page 4, row 16: In nearshore environments such as those sampled, high amounts of organic carbon and even copper may be present. Such compounds can influence results from the cadmium reduction method. In practice, high levels of carbon and copper can lead to underestimates of total nitrogen. It would be good if the authors could show that carbon and copper levels were low enough not to influence the analysis.

Page 5, rows 14-15: The cleaning intervals seem unnecessarily long. During the productive season, considerable fouling can happen in a matter of a few weeks. The authors should explain in greater detail how they made sure sensor drift (caused by fouling) did not affect the recorded data.

Page 6, rows 25-26: See comment for rows 14-15 on page 5.

Page 15, rows 3-4: It can be very much expected that shallow waters located between the 54th and 55th latitudes are very dynamic, on both diurnal and seasonal scales. By contrast, it would be interesting to know more about spatial variation in phenology on both scales. This kind of information disappears when all temporal variation is lumped into boxplots. When temporal variation is summarized like this, there's a high likelihood that the results from the different stations look uniform merely by chance. Besides, statistical inference seems not to have been used to assess whether the stations differed from or were similar with each other in terms of temporal variation. See also general comments.

---

## Author Comment (AC1) · 20 Mar 2019

We would like to thank the referee and the handling topical editor for their constructive and useful comments. Below you will find the detailed replies to all comments and in the supplement the revised version of the manuscript with tracked changes (Manuscript_revised.pdf).

Best regards

Markus Franz on behalf of the authors

Response to referee comment 1

General comments

[Figure]

A) In my opinion, the novelty of the study applies, at the most, to the southern Baltic. At least regarding the northern parts of the Baltic (Sweden, Finland, Estonia), corresponding data from shallow, coastal environments can be found in a large number of reports and publications. Albeit those data are often considered trivial and have thus not been highlighted and summarized as systematically as in the manuscript. In any case, the authors should know that these kind of data do exist, use them for comparison and refer to the sources accordingly. Instead of a case study, there would rather be need for a systematic review of data from shallow, coastal areas of the Baltic.

Reply: We fully agree that our study applies mainly to the southwestern Baltic Sea, which is why we tried to focus the manuscript on that area (e.g. title of the manuscript). It was not the intention to give conclusions in a context considering the entire Baltic Sea, but rather give local examples of strong environmental variabilities in shallow waters. This is also the reason why we compare our data only at a local scale. The fact that similar data from other regions of the Baltic exists, but were not found by the authors (as data might be hidden in grey literature or local databases) stresses even more the need for systematic publications of monitoring data. A systematic review of data from shallow, coastal areas of the entire Baltic Sea lies outside the scope of this manuscript.

Changes in the manuscript: A sentence was added to the introduction (page 1, line 28–29; page 2, line 1) addressing the accessibility of environmental data from monitoring programs.

B) Although the manuscript is supposed to be of descriptive nature, a proper synthesis of the findings is lacking. This would be especially important in a study without a specific theoretical framework or hypothesis.

Reply: Please see reply to point C.

C) One of the main points of the study is to highlight the fact that shallow, coastal environments are highly dynamic. However, this is hardly surprising. At least I would not

expect anything else than high diurnal and seasonal variation in most environmental variables from shallow, brackish waters located between the 54th and 55th latitudes. While the authors highlight the dynamism of the surveyed environmental type, they don't summarize its (diurnal and seasonal) dynamics in any kind of way. A spatial comparison of the dynamics and potential phenological differences among the stations would be a major improvement in the manuscript. This would be possible with the existing data.

Reply: The manuscript provides now a spatial comparison of the overall dynamics in temperature, salinity and dissolved oxygen concentration. The stations at the extremes of the spatial coverage (Stations 2 and 13) were chosen as examples for spatial and temporal differences, in order to keep the amount of plots acceptable.

Changes in the manuscript: A sentence was added to the data processing section, describing that data were plotted for two stations (page 7, line 5–6). A new paragraph was added to the data overview section (page 12, line 16–22; page 13, line 1–3) as well as to the conclusions section (page 16, line 6–19) to describe the results and give a synthesis of the findings. Figure 7 was added to depict the recorded data of Stations 2 and 13 (page 14).

D) In order to highlight the dynamic nature of the studied environments, they should be compared with something. First, the authors should refer to corresponding data from other shallow environments (see point A). More importantly, the set-up of the study should have included deeper waters further off the shore. Preferably, the study set-up should have incorporated coastal gradients (inshore-offshore) by each station. Although it is understandable that this kind of an approach would have required completely different resources, highlighting the dynamism of the studied environment without actually comparing it with anything is, in my opinion, a major issue.

Reply: As the reviewer is already indicating, extending the setup at each station to include a depth gradient would exceed the financial and logistic capacities of this monitor-

ing project. Furthermore, expanding the interpretation of the data further than already done (but consider the new version of the manuscript), would be out of the scope of the targeted publication. The overall aim is to give a detailed description of the dataset, allowing the reader an easy access and further use of the data. Additionally, we do not know of monitoring stations in the study area where temperature, salinity and dissolved oxygen concentration are all together recorded at a 10 min interval. Since the variability of the data is strongly related to the measurement interval, we did not compare our data to measurements taken on a daily or even monthly basis. Only for the data of dissolved inorganic nutrients, such a comparison was possible (see page 16, line 24–26).

Specific comments

Page 2, rows 12-15: Benthic communities will face not only low or high extremes but high variation in environmental conditions. In shallow areas, both sorts of extremes are very common.

Reply: The chosen formulation was a bit ambiguous. We were referring to the special case of coastal upwelling, not which conditions organisms are facing in general. The strong variability of environmental conditions in shallow water areas is mentioned in the following sentence.

Changes in the manuscript: The mentioned sentence was edited (page 2, line 17-18).

Page 2, rows 16-18: At least for the northern Baltic, there is already plenty of reports and scientific publications where this type of data can be found. However, those data may be considered too trivial to be summarized in their own right. In other words, the systematic summary presented in this manuscript can be considered useful (but not without references to and comparisons with data from other shallow, coastal areas of the Baltic).

Reply: Please see reply to point A and D.
Page 2, rows 23-25: As the purpose of the study is to show that shallow, nearshore habitats are particularly dynamic compared with deeper, off-shore habitats, depth gradients should absolutely be included in the sampling set-up (see also general comments).

Reply: Please see reply to point D.

Page 4, rows 4-9: Wading to the sampling stations can have had large effects on many of the measured variables because of sediment resuspension. The authors should be able to show that such effects did not take place. In my opinion, wading should absolutely be avoided, preferring other ways of approaching the stations.

Reply: The water samples have been taken from a water depth of 1 m. Approaching the stations in chest waders seems to us the least invasive method. Using a boat, besides practical and logistical issues, would clearly resuspend more sediment by its outboard motor compared to a person in waders. It is clear that we cannot exclude any effect by wading to the stations, but we think water movement by waves should keep this effect marginal.

Page 4, row 16: In nearshore environments such as those sampled, high amounts of organic carbon and even copper may be present. Such compounds can influence results from the cadmium reduction method. In practice, high levels of carbon and copper can lead to underestimates of total nitrogen. It would be good if the authors could show that carbon and copper levels were low enough not to influence the analysis.

Reply: The interference of copper and organic carbon to the cadmium reduction method are well described in the literature and therefore taken into account by the applied standard procedures, cited in table 2. Consequently, no measurements for copper and organic carbon concentrations were performed.

Page 5, rows 14-15: The cleaning intervals seem unnecessarily long. During the productive season, considerable fouling can happen in a matter of a few weeks. The

authors should explain in greater detail how they made sure sensor drift (caused by fouling) did not affect the recorded data.

Reply: Even under increased cleaning effort in summer and autumn, fouling could not be avoided completely. Therefore quality control procedures and in addition the visual inspection of the data were applied to identify measurements resulting from drifting sensors.

Changes in the manuscript: A sentence was added to clarify that fouling was an issue in some cases and data was therefore flagged accordingly (page 5, line 15–16).

Page 6, rows 25-26: See comment for rows 14-15 on page 5.

Reply: See reply to previous comment.

Changes in the manuscript: A more detailed description was added explaining the procedure of visual data inspection (page 6, line 24–25; page 7, line 1–2). In addition a sentence was added regarding the selection of threshold values for quality control procedures (page 6, line 22–23). Thresholds were chosen to be higher than the accuracy of the respective logger, to avoid misclassification of data that varied just as a result of the accuracy of the logger.

Page 15, rows 3-4: It can be very much expected that shallow waters located between the 54th and 55th latitudes are very dynamic, on both diurnal and seasonal scales. By contrast, it would be interesting to know more about spatial variation in phenology on both scales. This kind of information disappears when all temporal variation is lumped into boxplots. When temporal variation is summarized like this, there's a high likelihood that the results from the different stations look uniform merely by chance. Besides, statistical inference seems not to have been used to assess whether the stations differed from or were similar with each other in terms of temporal variation. See also general comments.

Reply: An exemplary spatial comparison of the dynamics in temperature, salinity and

dissolved oxygen concentration was added, but see reply to point C. The statistical analysis of the data would represent a major interpretation of the data, which is not intended for this type of publication.

Please also note the supplement to this comment:
https://www.earth-syst-sci-data-discuss.net/essd-2018-159/essd-2018-159-AC1-supplement.pdf

**Supplement:**

[revised manuscript text omitted]
. The applied thresholds were selected in order to guarantee the detection of erroneous sensor measurements that exceeded the accuracy of the method. Data that passed both tests were, in addition, visually inspected. Here, emphasis was put on erroneous readings resulting from biofouling of the sensors. Optical sensors typically show drift towards saturation. Conductivity sensors tend to steadily decrease, if the sensor is being fouled (Garel and Ferreira, 2015). Therefore,

the complete dataset for both sensor types were plotted and inspected. The measurements of DO concentration were checked for extended periods of full saturation, often visible as plateaus in the plots. For the salinity data, the visual inspection focused on continuous declines followed by immediate returns to measured values before the decrease started. Dates of read–outs and sensor cleaning were additionally used to support the identification of sensor drift. All data values were flagged according to applied quality checks (Table 4).

In order to exemplify spatial differences in overall variability as well as phenological differences among stations, entire datasets of temperature, salinity and dissolved oxygen concentration were plotted for Stations 2 and 13.

[revised manuscript text omitted]

---

## Referee Comment (RC2) · Anonymous Referee #2 · 9 May 2019

General comments:

Franz and colleagues provide environmental parameters, i.e. dissolved inorganic nutrient concentrations as well as temperature, salinity and dissolved oxygen, for parts of the German coast in the south-western Baltic Sea, over a two-year time frame. The manuscript is well written and structured, as is the provided data. The data is hosted on PANGEA and stored in a convenient to use and easily accessible format. All necessary information is provided as indicated in the manuscript. Overall, the manuscript and data can be a valid contribution to the field and seem of good quality. However, I do have some concerns that would need to be addressed before I can fully recommend the manuscript for publication. Please see these comments in the section below.

Specific comments:

[Figure]

Page 1, L 9-10: Most of local environmental monitoring in the Baltic Sea and elsewhere is not performed by large research vessels, and the potential scarcity of the data is very unlikely linked to the lack of accessibility by those vessels but rather by efforts to make available data public. Coastal data in the Baltic Sea is in fact not that scare. I recommend to rephrase this sentence.

P2, L2: I don't understand how local short-term and fine scale variability of environmental data can improve the predictions of climate change impacts. Please rephrase or elaborate.

P2, L11: The authors state that shallow areas are generally not affected by oxygen depletion. Especially in the Baltic Sea this is not true. Please see e.g. Conley et al. 2011 Environmental Science & Technology 45 (16), 6777-6783. Hypoxia in shallow coastal areas is rather common phenomenon, which is not only fueled by upwelling but rather by local sediment and physical conditions as well as respiratory processes due to excess organic content. This should be mentioned and addressed.

P2, L12 and following: When only referring to changing environmental conditions induced by upwelling events, this may be true. However, benthic and demersal species are also facing the opposite effects induced by climate related impacts, which the authors miss to report here, i.e. higher temperature and lower salinities. As climate related impacts and the usefulness of this study to understand them better are mentioned several times in the manuscript it would be important address this here as well and not only refer to upwelling.

P2, L18: Here and elsewhere (e.g. P1 L11, P15 L4) in the manuscript the authors make it sound like the study covers the south-western coast of the Baltic Sea. However, in fact the study is quite local and only covers a fraction of the German coast. I believe the authors should mention this more clearly besides in the specific material and methods part to avoid misleading information and generalization.

P4 L5 and L23: How did the person taking the water sample reach the sites where

loggers were deployed at 2.5 m depth wearing a chest wader? Or did she/he take the samples located away from the loggers? The sampling procedure is unclear for those cases. Please clarify.

P5, L14: As a reader I would like to know more about the irregular cleaning intervals (approximate time frames, degree of fouling and its potential impact in the measurements). As biofouling during peak summer season can have an impact on loggers within days this would be important information.

P6 , L21: How were the threshold values for the tests defined and based on what information criteria?

P11 and in general: The manuscript is presenting shallow coastal areas as dynamic and variable systems that can undergo rapid and large changes, which is true and important. However, what is clearly missing in the presentation of the data, and one the biggest issues in the current version, is the highlighting of the temporal variation within each site. There are only vague site-specific trends shown in figures 5 and 6. To fully appreciate the magnitude of short-term and site-specific variability, the sites would need to be plotted over time so that the reader gets an impression on what is happening at which scales. Interesting would for example be to see if there are differences in variability over seasons and in between years. How do diurnal, seasonal and annual differences in variation scale?

P12, L9: Instead of making the reader guess a trend over the box-plots, I would encourage the authors to provide a regression slope to highlight trends across stations. Also, is this a true geographical gradient following the salinity gradient of the Baltic Sea or is this rather because of the location of sites 10, 11 and 13, reflecting a local gradient towards the inner Lübeck Bight?

L11. Although this is a descriptive manuscript, some of the highlighted results would need to be put into perspective and be explained. What was this large change in salinity attributed to? How long did it last? Was it a specific upwelling event? Did this

coincide with a change in temperature and the other measured parameters? How do the authors explain such dramatic changes?

---

## Author Comment (AC2) · 5 Jun 2019

We would like to thank the referee for the constructive and useful comments. Below you will find the detailed replies to all comments and in the supplement the revised version of the manuscript with tracked changes (Manuscript_revised_1.pdf).

Best regards

Markus Franz on behalf of the authors

Response to referee comment 2

General comments

Franz and colleagues provide environmental parameters, i.e. dissolved inorganic nu-

trient concentrations as well as temperature, salinity and dissolved oxygen, for parts of the German coast in the south-western Baltic Sea, over a two-year time frame. The manuscript is well written and structured, as is the provided data. The data is hosted on PANGEA and stored in a convenient to use and easily accessible format. All necessary information is provided as indicated in the manuscript. Overall, the manuscript and data can be a valid contribution to the field and seem of good quality. However, I do have some concerns that would need to be addressed before I can fully recommend the manuscript for publication. Please see these comments in the section below.

Specific comments

Page 1, L 9-10: Most of local environmental monitoring in the Baltic Sea and elsewhere is not performed by large research vessels, and the potential scarcity of the data is very unlikely linked to the lack of accessibility by those vessels but rather by efforts to make available data public. Coastal data in the Baltic Sea is in fact not that scare. I recommend to rephrase this sentence.

Reply: We deleted this sentence from the abstract, since it was not contributing to summarize the study.

Changes in the manuscript: See changes on page 1, lines 9–10.

P2, L2: I don't understand how local short-term and fine scale variability of environmental data can improve the predictions of climate change impacts. Please rephrase or elaborate.

Reply: This sentence was already changed in response to referee 1.

Changes in the manuscript: See changes in response to referee #1 (blue highlights) on page 1, line 29 and page 2, lines 1–8.

P2, L11: The authors state that shallow areas are generally not affected by oxygen depletion. Especially in the Baltic Sea this is not true. Please see e.g. Conley et al. 2011 Environmental Science & Technology 45 (16), 6777-6783. Hypoxia in shallow

coastal areas is rather common phenomenon, which is not only fueled by upwelling but rather by local sediment and physical conditions as well as respiratory processes due to excess organic content. This should be mentioned and addressed.

Reply: The sentence was clarified and the respective reference was added.

Changes in the manuscript: See changes on page 2, lines 17–19.

P2, L12 and following: When only referring to changing environmental conditions induced by upwelling events, this may be true. However, benthic and demersal species are also facing the opposite effects induced by climate related impacts, which the authors miss to report here, i.e. higher temperature and lower salinities. As climate related impacts and the usefulness of this study to understand them better are mentioned several times in the manuscript it would be important address this here as well and not only refer to upwelling.

Reply: We added sentence to clarify that climate related impacts are occurring in these areas and that upwelling events might shift some drivers into the opposite direction, over short durations.

Changes in the manuscript: See changes on page 2, lines 22–24.

P2, L18: Here and elsewhere (e.g. P1 L11, P15 L4) in the manuscript the authors make it sound like the study covers the south-western coast of the Baltic Sea. However, in fact the study is quite local and only covers a fraction of the German coast. I believe the authors should mention this more clearly besides in the specific material and methods part to avoid misleading information and generalization.

Reply: The respective parts were changed to "the Baltic coast of Schleswig-Holstein, Germany", to make clear that the monitoring was performed at a rather local scale.

Changes in the manuscript: See changes on page 1, line 11, page 2, lines 27–28 and page 16, line 4.

P4 L5 and L23: How did the person taking the water sample reach the sites where loggers were deployed at 2.5 m depth wearing a chest wader? Or did she/he take the samples located away from the loggers? The sampling procedure is unclear for those cases. Please clarify.

Reply: The water samples for nutrient analysis were always taken at a depth of 1.2 m. In case stations were also serving as logger stations, the person walked into the direction of the logger station, to take the water sample as close as possible to the logger station. To clarify, the depth is mentioned again in the description.

Changes in the manuscript: See changes on page 4, line 8.

P5, L14: As a reader I would like to know more about the irregular cleaning intervals (approximate time frames, degree of fouling and its potential impact in the measurements). As biofouling during peak summer season can have an impact on loggers within days this would be important information.

Reply: Additional cleanings (besides during read–outs) were mostly done in the months June–October. Since the cleanings also required diving activities, weather conditions and staff availability did not allow repeated cleanings within weeks. Therefore, in some cases loggers were cleaned only once within this period. However, as the reviewer is indicating, fouling occurs within days in summer, an interval that would be impossible to maintain. To compensate for this, different quality control procedures were applied to the data. Among them, visual inspection of the entire dataset helped to identify suspicious data. This procedure was already described in more detail in response to referee #1.

Changes in the manuscript: See changes in response to referee #1 (blue highlights) on page 7, lines 3–10.

P6 , L21: How were the threshold values for the tests defined and based on what information criteria?

Reply: This question has been addressed in the response to referee #1: "Thresholds were chosen to be at least 2–times higher than the accuracy of the respective logger, to avoid misclassification of data that varied just as a result of the accuracy of the logger."

Changes in the manuscript: See changes in response to referee #1 (blue highlights) on page 5, lines 20–21 and on page 7, lines 5–8.

P11 and in general: The manuscript is presenting shallow coastal areas as dynamic and variable systems that can undergo rapid and large changes, which is true and important. However, what is clearly missing in the presentation of the data, and one the biggest issues in the current version, is the highlighting of the temporal variation within each site. There are only vague site-specific trends shown in figures 5 and 6. To fully appreciate the magnitude of short-term and site-specific variability, the sites would need to be plotted over time so that the reader gets an impression on what is happening at which scales. Interesting would for example be to see if there are differences in variability over seasons and in between years. How do diurnal, seasonal and annual differences in variation scale?

Reply: Since referee #1 raised a very similar point, a comparison of stations 2 and 13 was already included in the previous version manuscript. Temporal variations in temperature, salinity and dissolved oxygen concentration are described in the results section and discussed in the conclusions. The plot (Fig. 7) highlights seasonal fluctuations of the measured parameters and shows that site–specific differences were observed. To keep the number of plots acceptable, we exemplarily chose the two stations. In addition, a full interpretation of the obtained data is not intended for this type of publication (but see reply to last comment).

Changes in the manuscript: See changes in response to referee #1 (blue highlights) on page 12, lines 17–22; page 13, lines 1–4; page 14 (plot); page 16, lines 6–24.

P12, L9: Instead of making the reader guess a trend over the box-plots, I would encourage the authors to provide a regression slope to highlight trends across stations.

[Figure]

Also, is this a true geographical gradient following the salinity gradient of the Baltic Sea or is this rather because of the location of sites 10, 11 and 13, reflecting a local gradient towards the inner Lübeck Bight?

Reply: A linear regression between the straight–line distances from Station 2 to Station 13 (NW to SE) and salinity was applied. The resulting regression slope and the expected change in salinity over distance are now reported. Also, in the frame of the exemplary comparison between Stations 2 and 13, the driving forces of differences in salinity (geographical gradient or local factors) are discussed.

Changes in the manuscript: See changes on page 12, lines 9–10 and on page 16, lines 9–14.

L11. Although this is a descriptive manuscript, some of the highlighted results would need to be put into perspective and be explained. What was this large change in salinity attributed to? How long did it last? Was it a specific upwelling event? Did this coincide with a change in temperature and the other measured parameters? How do the authors explain such dramatic changes?

Reply: The strongest diurnal fluctuations (highlighted in Sect. 4.2) have now been discussed in the conclusions within the frame of the exemplary comparison of Stations 2 and 13. As all of the presented diurnal fluctuations were found in the data of Station 13, this appeared as the most natural way to discuss these results. The reviewer already highlighted that the manuscript should be rather of a descriptive nature and we think that an exemplary discussion of the results is the best tradeoff between a pure methodological description and an extensive interpretation of the entire dataset.

Changes in the manuscript: See changes on page 16, lines 6–24

Please also note the supplement to this comment:
https://www.earth-syst-sci-data-discuss.net/essd-2018-159/essd-2018-159-AC2-supplement.pdf

[Figure]

**Supplement:**

[revised manuscript text omitted]